# 3′UTR Deletion of *NONO* Leads to Corpus Callosum Anomaly, Left Ventricular Non-Compaction and Ebstein’s Anomaly in a Male Fetus

**DOI:** 10.3390/diagnostics12102354

**Published:** 2022-09-28

**Authors:** Maria Grazia Giuffrida, Marina Goldoni, Maria Luce Genovesi, Giovanna Carpentieri, Barbara Torres, Anca Daniela Deac, Serena Cecchetti, Anna Martinelli, Alessandro Vaisfeld, Elisabetta Flex, Laura Bernardini

**Affiliations:** 1Cytogenetics Unit, IRCCS Casa Sollievo della Sofferenza Foundation, Viale Cappuccini, snc, 71013 San Giovanni Rotondo, Italy; 2Department of Experimental Medicine, Sapienza University of Rome, 00185 Rome, Italy; 3Genetics and Rare Diseases Research Division, Ospedale Pediatrico Bambino Gesù, IRCCS, 00146 Rome, Italy; 4Department of Obstetrics and Gynaecology, AUSL Romagna, Infermi Hospital, 47923 Rimini, Italy; 5Core Facilities Confocal Microscopy Unit, Istituto Superiore di Sanità, 00161 Rome, Italy; 6Institute of Genomic Medicine, Catholic University School of Medicine, 00168 Rome, Italy; 7Department of Oncology and Molecular Medicine, Istituto Superiore di Sanità, 00161 Rome, Italy

**Keywords:** *NONO*, left ventricular non-compaction (LVNC), ebstein’s anomaly, corpus callosum agenesis, prenatal diagnosis

## Abstract

*NONO* (*Non-Pou Domain-Containing Octamer-Binding Protein*) gene maps on chromosome Xq13.1 and hemizygous loss-of-function nucleotide variants are associated with an emerging syndromic form of intellectual developmental disorder (MRXS34; MIM #300967), characterized by developmental delay, intellectual disability, poor language, dysmorphic facial features, and microcephaly. Structural brain malformation, such as corpus callosum and cerebellar abnormalities, and heart defects, in particular left ventricular non-compaction (LVNC), represent the most recurrent congenital malformations, recorded both in about 80% of patients, and can be considered the distinctive imaging findings of this disorder. We present on a further case of *NONO*-related disease; prenatally diagnosed in a fetus with complete corpus callosum agenesis; absence of septum pellucidum; pericallosal artery; LVNC and Ebstein’s anomaly. A high-resolution microarray analysis demonstrated the presence of a deletion affecting the *NONO* 3′UTR; leading to a marked hypoexpression of the gene and the complete absence of the protein in cultured amniocytes. This case expands the mutational spectrum of MRXS34, advises to evaluate *NONO* variants in pre- and postnatal diagnosis of subjects affected by LVNC and other heart defects, especially if associated with corpus callosum anomalies and confirm that CNVs (Copy Number Variants) represent a non-negligible cause of Mendelian disorders.

## 1. Introduction

Hemizygous loss-of-function variants of the *NONO* gene (*Non-Pou Domain-Containing Octamer-Binding Protein*, MIM * 300084), located on chromosome Xq13.1, have been recently described in association with syndromic intellectual developmental disorder-34 (MRXS34; MIM #300967), an X-linked recessive syndrome, characterized by delayed psychomotor development, intellectual disability with poor speech, dysmorphic facial features, macrocephaly, and mild structural brain malformation, including abnormalities of the corpus callosum and cerebellar hypoplasia in male patients [1]. In that paper, a functional characterization of *NONO* protein in *NONO*-deficient mice was performed suggesting its involvement in regulating RNA metabolism responsible for dendritic spine morphology and synaptic morphology at a cellular level [1]. After the description of the first three patients, a further clinical feature has emerged as recurrent in most of the patients. Indeed, in 2016 a subject affected by developmental delay and dysmorphisms associated with noncompaction cardiomyopathy (LVNC) was analysed by Whole Exome Sequencing (WES) and a novel de novo splice-site variant in *NONO* gene was found. The authors suggested that LVNC may be part of this newly described syndrome [2]. Effectively, since then, 73% of subjects affected by MRXS34 were reported showing cardiac anomalies, in particular LVNC, but also septal defects and right-sided lesions as Ebstein’s anomaly [3,4,5,6]. Heart defects were also disclosed as the peculiar clinical characteristic of a series of male fetuses with *NONO* variants detected by WES analysis [5,7]. Therefore, this characteristic associated with the neurodevelopmental phenotype could actually address the diagnosis of *NONO*-related disorder.

We report on a further male fetus referred to Copy Number Variants (CNVs) analysis due to a cardiac malformation and absence of cavum of septum pellucidum detected by II trimester ultrasound screening, and carrying a deletion involving the *NONO* 3′UTR, an intronic sequence of *BCYRN1* and entirely encompassing the nearby gene *ITGB1BP2*. The suspected functional role of the 3′ non-coding sequence alteration of the gene was established in the index case documenting the extremely low expression of *NONO* RNA and the complete absence of the protein in amniocytes. Therefore, this case expands the mutational spectrum of *NONO*, allows to better define the prenatal phenotype associated with *NONO*-related disorder and confirms that this gene is crucial in heart and brain development.

## 2. Materials and Methods

### 2.1. Case Report

A 41-year-old pregnant woman was referred at 22 weeks of gestation for genetic counseling due to an abnormal ultrasound with suspected corpus callosum agenesis and heart defect. The family history was unremarkable for chromosomal abnormalities, congenital anomalies, metabolic disorders or drug exposure before pregnancy. The couple has a healthy son of 7 months of age and a healthy daughter from a previous relationship. The first trimester screening for trisomies 21, 18 and 13 turned out at low risk and ultrasound screening at 13 + 2 weeks was unremarkable. The II trimester ultrasound screening (21 + 2) did not detect the cavum of septum pellucidum and suspected an atrial-ventricular canal defect. Two II level fetal ultrasonographies performed at 21 + 2 and 22 + 1 weeks and a fetal MRI performed at 21 + 4 weeks confirmed the absence of septum pellucidum and pericallosal artery with complete corpus callosum agenesis. Heart evaluation revealed cardiomegaly with wall hypertrophy and disproportion with a prevalence of the right heart chambers, wide tricuspid insufficiency and downward displacement of septal leaflet, suspecting Ebstein’s anomaly (Figure 1).

The other fetal structures appeared normal. After discussing the possible outcomes, the couple decided to terminate the pregnancy. Amniocentesis was however performed at 21 + 3 weeks, looking for a molecular diagnosis in view of future pregnancies. Post-mortem examination was not performed because of family denial.

### 2.2. Cell Cultures

Amniocytes from case index and from healthy donors were cultured in Chang C medium (FUJIFILM Irvine Scientific, Santa Ana, CA, USA) supplemented with L-glutamine (200 mM) and 1% penicillin-streptomycin, at 37 °C with 5% CO_2_.

### 2.3. CNVs Analyses

CNVs analysis on fetal DNA extracted from cultured amniocytes was performed using a genomic oligonucleotide-array (180K; Agilent Technologies Inc., Waldbronn, Germany) according to standard protocol with some modifications. In brief, fetal and reference DNA were heat-fragmented and labelled with Cy5-dCTP and Cy3-dCTP (Agilent Technologies Inc.), respectively. After washes, dried slides were immediately scanned on the Agilent scanner and analysed using Cytogenomics software (version 5.1.2.1; ADM-2 algorithm; release hg19) (Agilent Technologies Inc.). Called CNVs were represented by at least three consecutive probes, with an effective resolution of 75 Kb, and were classified based on the ACMG and ClinGen recommendations [8].

A further microarray analysis was performed using a SNP/oligo exonic microarray platform (XON Cytoscan chip; Thermo Fisher Scientific, Waltham, MA, USA) in order to better refine the breakpoints of the deleted fragment and in particular the proximal breakpoint involving the *NONO* sequence. This microarray consists of 6.55 M oligo and 300,000 SNP probes, covering the whole genome with a particular focus on about 7000 disease-relevant genes. The analysis was performed following the manufacturer’s suggestions. Intragenic CNVs were considered if covered by at least six probes. Data analysis was conducted using the Chromosome Analysis Suite software (ChAS; version 4.2.1; Thermo Fisher Scientific).

### 2.4. Real-Time PCR

Real-Time quantitative PCR (qPCR) for CNV segregation analysis and for cDNA quantification, in *NONO* expression experiment, was performed. cDNA was obtained by retrotranscription with random examers of 1µg RNA extracted from cultured amniocytes of the fetus and three normal controls according to the protocol of SuperScript IV (ThermoFisher Scientific). These analyses were carried out in triplicate on each sample using an ABI 7900 Sequence Detection System (Applied Biosystems, Foster City, CA, USA) and DNA-binding dye SYBR Green (Invitrogen Corporation, Carlsbad, CA, USA). For calculation of gene copy-number and gene expression the 2^−ΔΔCt^ comparative method was used [9], using *TERT* and *POLR2A*, respectively, as reference gene (See Appendix A for primers sequence).

### 2.5. Sanger Sequencing

The extent of the deletion and the exact location of the breakpoints were determined by a series of preliminary PCR amplifications using inward-facing primers selected based on the breakpoints mapping (see Appendix A for primers sequence), followed by Sanger sequencing, performed using the ABI BigDye Terminator Sequencing Kit (v3.1) and run on a 3130xl Genetic Analyzer (Applied Biosystems). Sequence electropherograms were analysed using ChromasPro (v1.7.5; Technelysium Pty Ltd., Brisbane, Australia).

### 2.6. Western Blot Analysis

To evaluate *NONO* expression level, cultured fetal and controls’ amniocytes were harvested and lysed using a RIPA buffer containing a phosphatase and protease inhibitors cocktail (Promega, Madison, WI, USA). For Western Blot analysis, equal amounts of total cell extracts were separated by 10% sodium dodecyl sulfate–polyacrylamide gel electrophoresis and transferred to nitrocellulose membranes (Bio-Rad, Hercules, CA, USA). Blots were then incubated with a rabbit polyclonal anti–*NONO* antibody (Cell Signaling, MA, USA). Membranes were re-probed with a mouse monoclonal anti-GAPDH (Santa Cruz Biotechnology, Dallas, TX, USA) to normalize protein content. Immunofluorescence assay

For immunofluorescence, patients’ and controls’ amniocytes were seeded at the density of 20 × 10^3^ in 24-well cluster plates onto 12-mm cover glasses. After 24 h of culture in complete medium, cells were fixed with 3% paraformaldehyde (30 min at 4 °C). After permeabilization with 0.5% TritonX-100 (10 min at room temperature), amniocytes were stained with anti-*NONO* rabbit polyclonal antibody (Cell Signaling), followed by the appropriate secondary antibody (Invitrogen) and DAPI. Confocal microscopy analyses were performed in three independent series of experiments on a Leica TCS SP2 AOBS apparatus (Leica Microsystems, Wetzlar, Germany) using excitation spectral laser lines at 405 and 488, using the confocal software (Leica Microsystems) and Photoshop CS5 (Adobe Systems, San Jose, CA, USA). Signals from different fluorescent probes were taken in sequential scanning mode, several fields of view (>100 cells) were analysed for each labeling condition, and representative results are shown.

## 3. Results

Chromosome microarray analysis disclosed a microdeletion on the X chromosome (Xq13.1) with a minimum size of 15 kb, spanning from 70,520,984 to 70,536,128 bp, and a maximum size of 35.84 kb, from 70,511,929 to 70,547,773 bp, based on the probe distribution of the microarray platform (Appendix A). The minimum deletion encompassed part of 3′UTR of *NONO*, the whole *ITGB1BP2* (Integrin, Beta-1, Binding Protein Of, 2, MIM * 300332) gene and an intronic sequence of *BCYRN1*, encoding a non-messenger RNA (https://genome.ucsc.edu/;hg19, accessed on 28 April 2022) and was classified as a variant of uncertain significance according to the American College of Medical Genetics and Genomics (ACMG) guidelines (8; Franklin, https://franklin.genoox.com/, accessed on 28 April 2020).

A further microarray analysis was performed using a SNP/oligo exonic microarray platform (XON Cytoscan chip; Thermo Fisher Scientific, Waltham, MA, USA) in order to better refine the breakpoints of the deleted fragment and in particular the proximal breakpoint involving the *NONO* sequence. This assay allowed to shift the initial breakpoint of *NONO* deletion at 70,519,969 bp, still inside the 3′UTR (Figure 2A). The quantitative Real Time-PCR (qPCR) on *BCYRN1* sequence performed on the parental samples revealed that the deletion was de novo (Figure 2B).

Sanger sequencing of the deletion junction fragment mapped the proximal and distal breakpoints at 70,519,973 bp and 70,541,997 bp, respectively, further confirming that the deletion involved almost the whole *NONO* 3′UTR, but not the coding sequence, being the sequence of the last coding exon entirely present in the sequenced breakpoints junction sequence. Sanger sequencing also ascertained the presence of a 4 bp insertion (5′-AAAA-3′) placed between the breakpoints and the presence of a 4 bp sequence homology between the proximal and distal boundary of the deletion and retained once in the deletion breakpoint-junction (Figure 2D).

The expression level of *NONO* in the fetus was evaluated on cDNA obtained from cultured amniocytes of the fetus by qPCR, revealing that *NONO* RNA, in the patient, was nine-fold reduced compared to healthy controls (Figure 3A). Moreover, both Western blot analysis and immunofluorescence assay performed on the fetal cultured amniocytes demonstrated the absence of *NONO* protein on the fetal amniocytes, whereas they documented a high protein level with a proper nuclear localization in amniocytes of healthy controls (Figure 3B,C).

## 4. Discussion and Conclusions

*NONO* (*Non-Pou Domain-Containing Octamer-Binding Protein*, MIM * 300084) gene is located on chromosome Xq13.1 and encodes a protein belonging to the highly conserved Drosophila behavior/human splicing (DBHS) protein family, which in mammals also includes *PSPC1* (*Paraspeckle Component 1*, MIM * 612408) and *SFPQ* (*Splicing Factor*, *Proline-And Glutamine-Rich*, MIM * 605199) [1], with which forms homo- and hetero-dimeric complexes. NONO protein is an RNA- and DNA-binding protein involved in DNA repairing, RNA splicing and stabilization and transcriptional activity [3]. For example, it regulates the circadian clock by repressing the transcriptional activity of the CLOCK-ARNTL/BMAL1 activator heterodimer. Pathway enrichment analysis suggested that *NONO* regulates genes mainly belonging to Gene Ontology categories related to synaptic functions. Finally, Mircsof and colleague documented that *NONO* plays a critical role in the regulation of synaptic RNAs and GPHN/gephyrin scaffold structure, through the regulation of *GABRA2* transcript [1]. These data directly correlate the molecular and cellular functions of this gene with the phenotype associated to its impairment, characterized by global developmental delay and/or intellectual disability associated with central nervous system (CNS) malformations, in particular corpus callosum hypoplasia/dysplasia/thickness, which were recorded in 8/10 (80%) patients where CNS was evaluated [1,2,3,4,5]. Therefore, the corpus callosum hypoplasia detected in the fetus represents a recurrent instrumental prenatal finding of *NONO* syndrome.

Less clear is the involvement of *NONO* in heart defects that has been confirmed to be present in the most of *NONO*-patients. From the first description of a subjects with a pathogenic variant of *NONO* presenting with LVNC [2], this cardiac defect was reported in the other 12 male patients, making this clinical feature as a distinctive sign of this disorder [2,3,4,5,6,7]. Based on literature review, LVNC was isolated in two cases (2/13; 15%), whereas was reported in association with other structural heart defects in the others (11/13; 85%). Septal anomalies were the most recurrent associated cardiac defect, in particular a ventricular septal defect (VSD) was recorded in eight of these cases (8/11; 73%) and an atrial septal defect in 4 (4/11; 36%). VSD represented the most associated recurrent cardiac feature also considering only unrelated cases (nine families) (5/9; 56%). LVNC is a rare structural heart defect due to myocardial morphogenesis interruption occurring between the 5th and 8th weeks of embryonic life, that prevents the trabecular regression and muscle tissue compaction. It results in a thickened myocardium with the two layers consisting of compacted and non-compacted myocardium and characterized by prominent trabeculae and deep intratrabecular recesses [10]. The exact cardiac phenotype was not replicated in the *NONO* deficient mouse model (NONO-KO) [11]. Moreover, male NONO-KO mice showed smaller hearts compared to the controls. The authors demonstrated that this heart morphology is associated with a defect in DNA replication, cell cycle, cell division and migration of cardiac fibroblasts, highlighting that *NONO* plays a role in heart development [11]. In the fetus herein described, ultrasound screening revealed the presence of several cardiac malformations recorded in *NONO* patients, as LVNC, apical ventricular defect and Ebstein’s anomaly that, nevertheless, are described often in co-morbidity [12]. Therefore, the current case brings further evidence that *NONO*-related phenotype is distinctly characterized by the presence of congenital heart anomalies. LVNC is considered a condition with increased risk of sudden cardiac death especially when associated with sport exercise, with the outcome likely dependent on specific echocardiographic characteristics, such as thickness value and ratio of non-compacted to compacted layer. Ebstein’s anomaly, a congenital defect of the tricuspid valve downward displaced into the right ventricle, is associated with decreased survival due to biventricular failure [13]. Therefore, the high recurrence of heart malformations in addition to ID and SNC anomalies should be taken into account in clinical management of MRXS34 patients.

The present case expanded *NONO* mutational spectrum, as it carries a genomic loss of about 22 kb involving only the 3′UTR of the gene, keeping the coding sequence intact, as demonstrated by the breakpoints junction sequencing. So far, most of the reported variants are located widespread along the sequence of the gene, apparently without mutational hotspots, and leading to the complete absence of the protein [1,3]. Therefore, in order to assess the pathogenicity of the genomic loss, the expression level of the *NONO* transcript and protein on cultured fetus’ amniocytes was examined, documenting a strong reduction of mRNA level, being near zero, and the complete absence of *NONO* protein (Figure 3). The 3′UTR represents the site in which post-transcriptional control is carried out and the functional consequences of variants only involving a portion of 3′UTR region are unpredictable. Numerous pathogenic variants mapping in 3′UTR have been reported and their examples are increasing due to the application of Whole Genome Sequencing (WGS) analysis [14]. Generally, these variants interfere with the binding of specific microRNA or mRNA binding proteins involved on the gene expression control, determining an aberrant mRNA stability and translation efficiency and causing an altered protein level synthesis or mRNA localization, whereas the complete ablation of the gene product is quite unexpected [15].

We speculate that, in this case, the deletion of a substantial part of the 3′UTR of *NONO* gene could preclude the endonucleolitic cut and poliA tail polymerization, preventing correct *NONO* mRNA processing and thus affecting its stability. This hypothesis is supported by the observation that in the *NONO* 3′UTR there are two putative PAS (PolyAdenylation Signal) sequences that are both included in the deleted region (Figure 1C).

The exact mapping and sequencing of the breakpoints junction fragment led us to speculate on the underlying mechanism of deletion formation. As showed in Figure 1D, both the boundaries of breakpoints carried an identical 4 bp sequence and was also demonstrated that this sequence is present once in the junction fragment together with the presence of an inserted 4 bp sequence of unknown origin. Therefore, we could hypothesise that a mechanism of serial fork stalling template switching/microhomology-mediated break induced replication (FoSTeS/MMBIR) was involved, as described by Hijazi and colleagues [16].

In conclusion, this case confirms the utility of CNVs analysis also for diagnosing Mendelian diseases and expands the mutational spectrum of MRXS34, highlighting that the deletion of *NONO* 3′UTR leads to the complete absence of the protein. Moreover, this further case of prenatally assessed *NONO* variant indicates that this gene should be taken into consideration in the genetic study of fetuses with a cardiac defect, particularly with LNVC and Epstein anomaly.

## Figures and Tables

**Figure 1 diagnostics-12-02354-f001:**
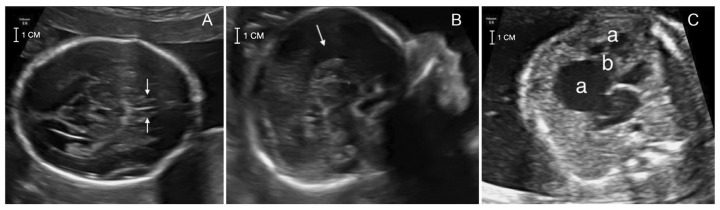
The II level ultrasound screening at 22 + 1 showing: (**A**) the absence of cavum of septum pellucidum (white arrows); (**B**) the corpus callosum agenesis (white arrow); (**C**) the heart anomaly with the prevalence of the right heart chambers (**a**) and displacement of the tricuspid valve structure (**b**), consistent with Ebstein’s anomaly.

**Figure 2 diagnostics-12-02354-f002:**
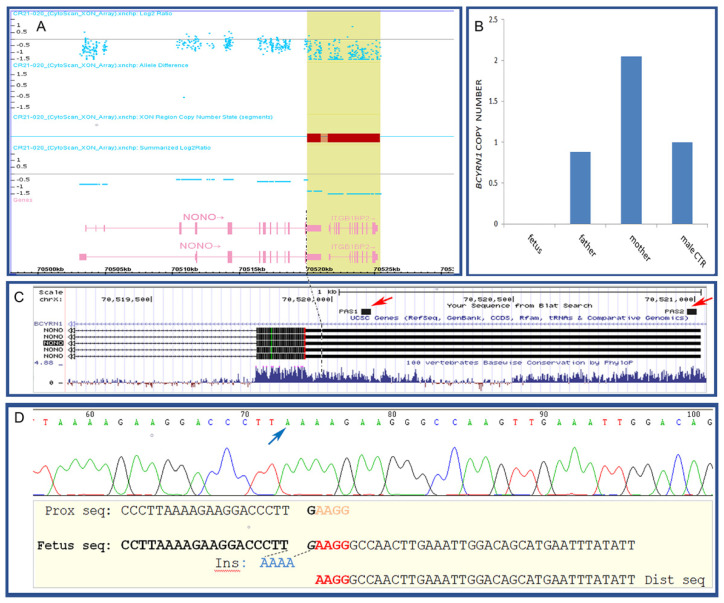
(**A**) the microarray analysis using an exonic-array platform (XON Cytoscan; Thermo Fisher Scientific) allowed to exclude the involvement of *NONO* coding sequence from the deleted segment (highlighted area). (**B**) the deletion originated de novo, as demonstrated by the histogram of Real-Time PCR performed on the fetus, the parental DNA and a normal male control (CTR) used as reference. (**C**) the proximal breakpoint mapped within the 3′UTR region (dotted line), based on the UCSC Genome Browser (https://genome.ucsc.edu/; release hg19, accessed on 29 April 2022). This image also represents the conservation level of the 3′UTR deleted region and the PASs (PolyAdenilation Signals) (arrows), both enclosed within the deleted segment. (**D**) Breakpoints junction sequence obtained by Sanger sequencing and imaged with Chromas (https://chromas.software.informer.com/2.5/, accessed on 29 April 2022). Below the reported breakpoints junction sequence showing 4 bp microhomology along the breakpoints boundaries (black bars), that is present once in the breakpoints junction sequence. On the left of this microhomology, the breakpoints junction sequence shows the inserted sequence AAAA of unknown origin (arrow).

**Figure 3 diagnostics-12-02354-f003:**
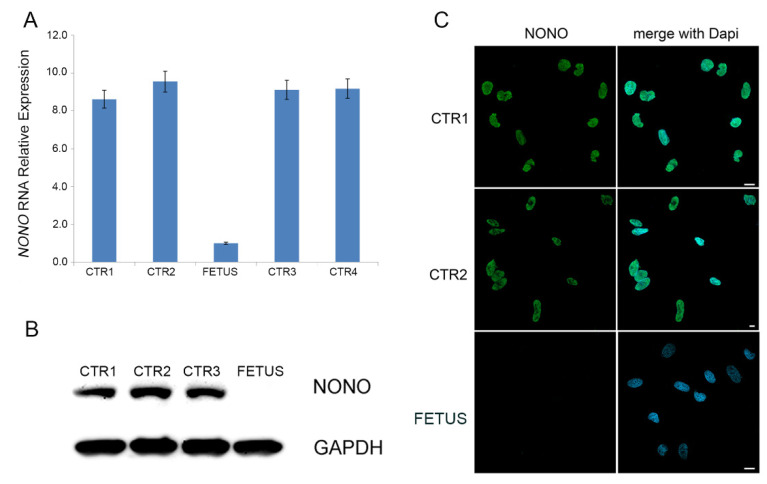
(**A**) Histogram showing about nine-fold decreased expression of *NONO* in fetal cultured amniocytes compared to four distinct control cells. Data were obtained from two independent experiments. (**B**) The extremely reduction of *NONO* RNA amount is reflected in the complete absence of the protein on patient cells compare to control cells (CTR), as showed by Western blot (WB) analysis. Representative blot of three independent experiments is shown. Equal amount of cell lysates was resolved by 10% polyacrylamide gel electrophoresis. Membranes were probed with anti-*NONO* antibody and signals were normalized using GAPDH as internal control. (**C**) Confocal microscopy analysis confirmed absence of the *NONO* protein in patients’ cells compared to controls’ amniocytes that showed high level of *NONO* protein and its proper nuclear localization. Scale bar represents 10 μm.

## Data Availability

The data presented in this study are openly available at DECIPHER; https://www.deciphergenomics.org/Case n.427944, accessed on 2 March 2021.

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
