# Peer review of "3′UTR Deletion of NONO Leads to Corpus Callosum Anomaly, Left Ventricular Non-Compaction and Ebstein’s Anomaly in a Male Fetus"

_diagnostics, 2022, doi:10.3390/diagnostics12102354_

Round 1
Reviewer 1 Report
I found it interesting that NONO is involved not only in proper development of central nervous system, but also in heart development. It appears that the deficiency of NONO occurs mainly due to de novo mutations. If this is the case, nearly all patients must be males as the title of this paper sugests. However, this is not clear in DISCUSSION. Are the 12 cardiac defect patients on line 237 all males? Did you confirm sex-dependency of phenotypes in NONO-KO in line 249-252?
Figure 1 and supplementary Figure 1 cannot be clearly seen. I hope these will be replaced by clearer ones.
Author Response
we thank the Reviewer for this observation. According to this suggestion, in the present version of the manuscript we clarified that the MSRX34 is a recessive X-linked condition (see Introduction, line 51) and that therefore all the patients described so far and the KO-mice showing the cardiac defects were males (see Introduction, line 54 and Discussion, line 240 and line 252).
Moreover, Figure 1 and Supplementary figure were replaced, although the quality of original pictures of Figure 1 did not allow to substantially improve it. Anyway, we added some details (see Figure 1C) and we hope that the present version answers your request.
Reviewer 2 Report
The article is well written and no need for changes.
Author Response
Thank you for your kind review.
No reply requested
Reviewer 3 Report
The authors presented a high-resolution microarray analysis of a case of NONO-related disease. The paper is well organized and the contents are interesting. One missing is the scale bar in the ultrasound images in Fig. 1.
Author Response
according to this useful observation, we added to the Figure 1 several details in order to make clearer the cardiac structures described and the scale was added.
We hope that this improvement satisfies the Reviewer request concerning the Methods.
Reviewer 4 Report
The article is interesting and well written.
I suggest to improve it following CARE checklist for case report - guidelines (https://www.care-statement.org/checklist) that could be useful for this and for your next articles.
Please consider the following reference when speaking about LVNC and Ebstein abnormalities (https://www.ncbi.nlm.nih.gov/pmc/articles/PMC8436685/), since these could be conditions recognizable through echocardiography that could influence a possibile future sport eligibility for those kids
Author Response
We thank the Reviewer for suggesting following the CARE checklist. We checked the format and to our opinion the case report appears to be as conform as can be, considering the type of case described and available information.
Moreover, we mentioned in Discussion the reference suggested (see lines 260-266). On this regard, please notice that a sentence of Discussion was moved upward (from lines 257-258 to lines 235-236) because in the new arrangement of Discussion the current position seemed more appropriate.